# The Conserved MAP Kinase MpkB Regulates Development and Sporulation without Affecting Aflatoxin Biosynthesis in *Aspergillus flavus*

**DOI:** 10.3390/jof6040289

**Published:** 2020-11-16

**Authors:** Sang-Cheol Jun, Jong-Hwa Kim, Kap-Hoon Han

**Affiliations:** Department of Pharmaceutical Engineering, Woosuk University, Wanju 55338, Korea; scjun@woosuk.ac.kr

**Keywords:** *Aspergillus flavus*, MAPK, *mpkB*, sclerotia, conidiation, aflatoxin

## Abstract

In eukaryotes, the MAP kinase signaling pathway plays pivotal roles in regulating the expression of genes required for growth, development, and stress response. Here, we deleted the *mpkB* gene (AFLA_034170), an ortholog of the *Saccharomyces cerevisiae FUS3* gene, to characterize its function in *Aspergillus flavus*, a cosmopolitan, pathogenic, and aflatoxin-producing fungus. Previous studies revealed that MpkB positively regulates sexual and asexual differentiation in *Aspergillus nidulans*. In *A. flavus*, *mpkB* deletion resulted in an approximately 60% reduction in conidia production compared to the wild type without mycelial growth defects. Moreover, the mutant produced immature and abnormal conidiophores exhibiting vesicular dome-immaturity in the conidiophore head, decreased phialide numbers, and very short stalks. Interestingly, the Δ*mpkB* mutant could not produce sclerotia but produced aflatoxin B1 normally. Taken together, these results suggest that the *A. flavus* MpkB MAP kinase positively regulates conidiation and sclerotia formation but is not involved in the production of secondary metabolites such as aflatoxin B1.

## 1. Introduction

*Aspergillus* section *Flavi* consists of 27 species, and *Aspergillus flavus* is a representative species that shows the characteristics of the section *Flavi* well [1,2]. The fungi that belong to this section exhibit a very high genetic diversity, which not only results in morphological differences but also variations in secondary metabolite production [3,4]. *A. flavus* is a cosmopolitan saprophytic soil fungus that infects grain (e.g., maize, and wheat), peanuts, and fruit crops, causing the crops to rot and damaging their productivity [5,6]. Importantly, *A. flavus* and *Aspergillus parasiticus* are known to produce mycotoxins including aflatoxin B1 and B2, which are considered the most potent carcinogens in nature [7,8,9]. Consumption of aflatoxin-contaminated crops and feeds by humans or livestock can lead to liver disease, liver cancer, acute physiological disorders, and death [9,10]. Furthermore, *A. flavus* is an opportunistic pathogenic fungus that causes invasive or noninvasive aspergillosis in humans and animals along with *Aspergillus fumigatus* [11,12].

*A. flavus* is mostly propagated via the production of asexual spores (conidia), and its mycelial cells are known to form sclerotia (i.e., a differentiated tissue) to overcome harsh environmental conditions or to withstand the winter season [13]. Genomic analyses have revealed that both *A. flavus* and *A. parasiticus* possess either a *MAT1-1* or *MAT1-2* allele, two idiomorphic *MAT* loci. *A. flavus* and *A. parasiticus* have been predicted to be genetically and physiologically capable of heterothallic fertility or parasexual recombination, as they conserve a variety of DNA fingerprint types and intraspecies vegetative compatibility groups [3,14]. Additionally, Horn et al. [15] confirmed the sexual development of *A. flavus* (teleomorph: *Petromyces flavus*) in cereal agar medium co-inoculated with different mating type strains for 6 to 11 months. During long-term experimental conditions, *A. flavus* produced sclerotia with a very hard and thick layer structure (sclerenchyma), which produced ascocarps inside the differentiated structure and produced asci, containing ascospores in the ascocarp.

Once the sequence of the *Aspergillus* genome became available, various genes related to the developmental processes of fungi and the production of secondary metabolites were identified. For example, the VeA protein of *A. nidulans* acts as a negative regulator of conidiation and a positive regulator of sexual development and secondary metabolism [16,17]. Moreover, the LaeA protein is known to be an important positive regulator of secondary metabolite production [18]. These two proteins are known to regulate conidiation, ascosporulation, and secondary metabolism in fungi by forming complexes with VelB and VosA, which are other members of the velvet protein family in *A. nidulans* [19,20].

In *A. flavus*, LaeA and VeA ortholog proteins were also characterized as important positive regulators of aflatoxin and sclerotia production [19,21]. The deletion of the *laeA* gene resulted in the loss of sclerogenicity and the expression of the *aflR* gene, which encodes the aflatoxin-specific transcription factor, thereby blocking aflatoxin production. Overexpression of *laeA*, on the other hand, was found to increase the production of sclerotia and aflatoxin. Furthermore, a *laeA* mutant of *A. flavus* exhibited abnormal phenotypes including a reduction in conidiation and loss of pigmentation on the bottom of the agar medium plate. Moreover, the expression of *veA* in the *laeA* deletion mutant was substantially upregulated, suggesting that the expression of *veA* is negatively regulated by the LaeA protein [21,22]. The *veA* gene of *A. flavus* and *A. parasiticus* is known to be essential for the production of mycotoxins such as cyclopiazonic acid, aflatrem, and aflatoxin, as well as for the formation of sclerotia. Similar to the *laeA* mutant, *veA* deletion in two fungi also repressed the expression of *aflR* [23,24]. Additionally, the LaeA and VeA proteins in other *Aspergillus* fungi such as *A. nidulans* and *A. fumigatus* were found to be related to the production of secondary metabolites and pathogenicity [18,25,26].

Furthermore, the NsdC and NsdD proteins, which are known transcription factors required for sexual development in *A. nidulans*, were found to be involved in a variety of differentiation processes in *A. flavus* including conidiophore development, conidiation, aflatoxin biosynthesis, and sclerotia production. *nsdC* and *nsdD* knock-out mutants in *A. flavus* do not produce sclerotia and exhibit abnormal conidiophores and fewer conidia compared to their wild-type counterparts [27,28,29,30].

The Fus3/Kss1 MAP kinase plays a pivotal role in the mating signal transduction pathway of *S. cerevisiae* and activates key transcription factors and regulators associated with sexual development such as Ste12, Far1, and Cdc24 [31,32,33]. The *A. nidulans* MpkB, an ortholog of the *S. cerevisiae* Fus3/Kss1 MAP kinase, is essential for the formation of cleistothecia and is also known to regulate hyphal anastomosis, post-karyogamy processes, and secondary metabolite biosynthesis [34,35,36]. Additionally, *A. nidulans* MpkB MAP kinase can directly interact with LaeA, VeA, and VosA and is known to phosphorylate VeA protein. MpkB forms a pheromone module with the upstream kinases MkkB and SteC, as well as the adapter protein SteD. The MpkB pheromone module is located on the membrane but moves across the cytoplasm to the nuclear envelope when the module is activated by extracellular stimulation. Afterward, the phosphorylated MpkB separates from the module and enters the nucleus alone. Activated MpkB phosphorylates VeA in the nucleus and was confirmed to directly bind to the C2H2-Zn finger transcription factor SteA [37]. Additionally, the scaffold protein HamE was shown to directly bind to the MpkB kinase module and act as a positive phosphorylation regulator of the module [38]. Moreover, three *A. flavus* kinases and SteD adapter proteins were physically linked to form a tetrameric complex, which had similar functions to the MpkB pheromone module of *A. nidulans*. However, unlike in *A. nidulans*, HamE (a known positive phosphorylation regulator of the pheromone module) was not physically linked to the tetrameric complex. Nevertheless, the MpkB pheromone module of *A. flavus* plays a critical role in the asexual sporulation, sexual sclerotia formation, and aflatoxin B1 production [39].

To investigate the factors that govern the developmental stages and secondary metabolite production of *A. flavus*, several mutants of the *mpkB* MAP kinase gene were constructed via knockout and overexpression cassettes. Through this approach, our study determined that the function of the *A. flavus* MpkB MAP kinase was associated with conidiation, conidiophore morphological development and sclerotial production. Loss of the *A. flavus mpkB* gene resulted in conidiophore morphological abnormalities, reduced conidiation, and blocked sclerotia production but did not affect aflatoxin B1 biosynthesis.

## 2. Materials and Methods

### 2.1. Strains and Growth Conditions

*A. flavus* strain NRRL3357-5 (*pyrG*^−^) [40] was used as a host strain for the integration of manipulated genes (Table 1). All *A. flavus* strains used in this study were cultured on complete medium (CM), potato dextrose agar (PDA), potato dextrose broth (PDB), and minimal medium (MM). CM and MM were prepared with some minor modification based on Pontecorovo et al. [41] and Käfer [42]. Minimal salt solution (MS) was pre-made as 20× and added to the medium after sterilization. The ingredients of MS were almost identical to those of Käfer [42], except for some trace elements (boric acid and cobalt chloride were not included). Uridine and uracil (final 10 mM each) were added to the medium whenever necessary. In *A. nidulans*, adding certain amounts of salts (e.g., 0.6 M KCl) promoted asexual development and inhibited sexual development [43]. In order to investigate the change of conidiation in *A. flavus*, the strains were cultured by adding 0.6 M potassium chloride to CM agar medium. All strains were grown on solid media at 30 °C or shaken in liquid media at 250 rpm at 30 °C. For RNA preparation, asexual reproductive development conditions were induced on the MM plate by spreading mycelial balls that had been grown in CM broth for 16 h at 30 °C. To mimic sexual development-inducing conditions (hereinafter referred to as “sexual induction”), the 16-h-grown mycelial balls were spread onto MM plates, after which the plates were tightly sealed with parafilm and incubated for 24 h in the dark. The plates were further incubated for a predetermined amount of time after unsealing [27].

### 2.2. Construction of Deletion and Overexpression Mutants

The *A. nidulans mpkB* gene ortholog (AFLA_034170) in *A. flavus* was identified via an in silico gene similarity search of the Aspergillus Genome Database (http://www.aspgd.org/). An *A. flavus mpkB* deletion strain was created by deletion/replacement of the entire *mpkB* transcriptional unit (from −935 to +1288 nt) at the resident *mpkB* locus on chromosome II. A double-joint PCR approach was used to make a deletion construct according to the protocol developed by Yu et al. [44]. The *A. fumigatus pyrG* selectable marker gene (*Afu_pyrG*) was amplified from genomic DNA with primers AfupyrGF1 and AfupyrGR1 (Appendix A). The upstream 1004 bp and downstream 880 bp genomic fragments flanking the *mpkB* transcriptional unit were amplified using primers that introduce 27-nt overlapping extensions with the selectable marker *Afu_pyrG*. The 5′-flank was amplified with primers MpkBD5F and MpkBD5R. Likewise, the 3′-flank was amplified using primers MpkBD3F and MpkBD3R (Appendix A). PCR fragments of *mpkB* and an *Afu_pyrG* amplicon were then used to conduct a double-joint PCR for the construction of the deletion cassette. The final fusion product was directly transformed into the *A. flavus* NRRL3357-5 (*pyrG*^−^) recipient strain. Transformants were recovered on the selective media, and the deletion of the *mpkB* transcription unit was confirmed by Southern blot and Northern blot analyses in at least 10 individual isolates. In order to overexpress the *mpkB* gene using an inducible promoter, we constructed an overexpression vector that fused the *A. nidulans niiA* promoter, *niiA*(p), followed by the *mpkB* open reading frame (ORF). The overexpression strain of *mpkB* was constructed by transforming the NRRL3357-5 strain with the *niiA*(p)*::mpkB* fusion plasmid pNII-mpkB. The pNII-mpkB vector harbored the *Afu_pyrG* gene for transformant selection. *pyrG*^+^ transformants were analyzed using PCR and Northern blots to confirm that the plasmid had been integrated and the *mpkB* gene was overexpressed. The induction of the *niiA* promoter was performed as described previously [45]. To generate a complemented *mpkB* strain, pPTR-mpkB was constructed by amplifying an approximately 3.7 kb region of NRRL3357 genomic DNA representing 1.3 kb of the *mpkB* coding region sequence using the mpkB5NEST and mpkB3NEST primer set (Appendix A). This PCR amplicon was ligated into the pPTR1 vector (TaKaRa, Japan) harboring the pyrithiamine resistance gene for the selection of transformants. The pPTR-mpkB plasmid was transformed into the *mpkB* deletion strain (SCWS1.11) and the transformants were screened with MM or uridine- and uracil-containing MM plates supplemented with 0.1 μg/mL pyrithiamine and confirmed through PCR (Appendix A).

### 2.3. Conidial and Sclerotial Production Analyses

A conidial suspension aliquot (10^−3^) was seeded at the center of each PDA or CM agar plate with the addition of uridine and uracil. To quantitatively compare conidia and sclerotia production, the aliquots were cultured in the dark at 30 °C for 4 days (conidial production) or 6 days (sclerotial production) in triplicate. At the end of the growth period, the conidia were washed off from the agar plates using 0.08% Triton X-100 solution and counted with a hemacytometer (conidial production) or photographed (sclerotial production).

### 2.4. Microscopy

Conidiophore morphology was examined from colonies growing on CM agar and PDA blocks mounted on glass slides or directly grown on plates using an Olympus BH2 light microscope.

### 2.5. Nucleic Acid Extraction and Northern Blot Analysis

Fungal genomic DNA for Southern blot analysis and PCR amplification was prepared from mycelia after surface culture incubation for 24 h at 30 °C in CM broth. The mycelial mat was then collected and squeeze-dried, after which genomic DNA was isolated as described by Yu et al. [44]. Total RNA was isolated from vegetative and/or induced mycelial mats using the TRIzol reagent (Invitrogen, Carlsbad, CA, USA). All nucleic acids were manipulated according to standard procedures [46]. Northern blots were conducted by submitting RNA to electrophoresis in a 1.0% formaldehyde gel, after which it was transferred to Hybond-N^+^ membranes (Amersham Bioscience, Amersham, Buckinghamshire, UK) via the capillary transfer blotting method. The probes were obtained through PCR using the primers described in Appendix A and labeled with the Random Primer DNA Labeling Kit Ver. 2 (TaKaRa Bio, Shiga, Japan) following the procedures recommended by the supplier.

### 2.6. Aflatoxin Extraction and Analysis via Thin-Layer Chromatography (TLC)

For aflatoxin analysis, conidia of *A. flavus* strains were inoculated into test tubes containing PDB media and incubated in stationary conditions at 30 °C for 3 days in the dark. Aflatoxin was extracted from the stationary cultures by adding 2 mL CHCl_3_ directly to the 3 mL culture, vortexing thoroughly, and allowing the mixtures to stand for 5 min at room temperature before collecting the organic phase and centrifuging at 14,000 rpm for 2–3 min to remove residual aqueous material. Finally, 500 μL of organic phases were collected and completely dried at room temperature, then resuspended in 50 μL of CHCl_3_. Thirty microliters of this solution were separated in a developing solvent containing acetone and chloroform (1:9 by volume) on silica gel TLC plates, then exposed to long-wave UV (365 nm) light.

## 3. Results

### 3.1. Generation of A. flavus mpkB Deletion Strains

The genomic DNA sequence of the *mpkB* ortholog (AFLA_034170) was found to be 1529 bp including the UTR site, had 5 introns, and its encoding protein was estimated to consist of 354 amino acids. The amino acid sequence similarity between *A. nidulans* MpkB and *A. flavus* MpkB was 99%. The deletion mutant of *A. flavus mpkB* was obtained by transformation of the *A. flavus* host strain NRRL3357-5 using a 3394 bp cassette and the *A. fumigatus pyrG* gene as a selection marker (Figure 1a). Eight putative *mpkB* deletion transformants were selected via diagnostic PCR analysis using the mpkB5NEST and mpkB3NEST primers. Southern blot analysis of these strains confirmed that all eight transformants lost the *mpkB* gene, which was substituted by the *Afu_pyrG* marker (Table 1). Southern blot analysis of the host strain genomic DNA was digested with *EcoR*V with the *mpkB* probe and exhibited a band of approximately 2.9 kb, whereas the Δ*mpkB* mutants showed no band due to the absence of the *mpkB*. In contrast, Southern blot analysis with the *Afu_pyrG* probe resulted in a band of approximately 4.3 kb in the Δ*mpkB* mutants, yet no band was observed in the host strain (Figure 1b). After confirming that the phenotypes of all eight Δ*mpkB* mutants were equal, we selected the SCWS1.11 Δ*mpkB* strain for subsequent experiments.

### 3.2. Deletion of mpkB Affects Conidiation and Sclerotia Production But Not Growth

The deletion of *mpkB* showed no difference in the radial colony growth or hyphal growth between wild-type and the Δ*mpkB* mutant cultured on CM agar at 30 °C for 4 days (Figure 2a). However, when the conidiation of wild-type and Δ*mpkB* mutant were compared after culturing in CM agar, an approximately 60% reduction in the Δ*mpkB* mutant conidia was observed. Given that the Δ*mpkB* mutant showed a decreased conidiation phenotype in CM agar, we verified whether the 0.6 M KCl condition, which promotes conidiation, could suppress the Δ*mpkB* mutant phenotype. As a result, the mutant grown on the CM with 0.6 M KCl exhibited a 30% reduction in conidia production compared to the wild type, suggesting that MpkB is essential for normal sporulation regardless of the environmental conditions, although the salt condition restored conidiation by 50%. Additionally, this phenotype was completely recovered from the *mpkB* complemental strain, indicating that the phenotype was attributable to the deletion of the *mpkB* gene (Figure 2b). Similar results were observed when identical experiments were performed in PDA media.

Since *mpkB* is a homolog of the yeast *F**US3* gene, we assumed that the loss of *mpkB* affects sclerotia formation in *A. flavus*, because sclerotia were identified as sexual structures in mated *A. flavus* [47]. As expected, the production of sclerotia was completely blocked in the Δ*mpkB* mutant. When the wild-type and Δ*mpkB* mutants were inoculated on the CM agar or PDA plate and incubated at 30 °C for 6 days under dark conditions, no production of sclerotia was observed in the Δ*mpkB* mutant (Figure 3). The inoculated plates were incubated for a longer period (maximum 15 days) under the same conditions, but no further production of sclerotia was observed in the Δ*mpkB* mutant. On the other hand, sclerotia production in the *mpkB* complemental strain reached the same level as that of the wild type (Figure 3).

### 3.3. Deletion of mpkB Caused Markedly Aberrant Conidiophore Morphology

Microscopic observations revealed abnormalities in the conidiophore morphology of the Δ*mpkB* mutant. The mutant had fewer conidial heads than the wild type, and the shape of the conidial heads was abnormal (Figure 4a,b). Additionally, the morphology of the vesicle dome isolated from the stalk tip did not have a clear shape and the amount of metulae was remarkably lower than that of the wild type. The size and shape of phialide differentiated from metulae was also not uniform, and the conidial head was far less dense (Figure 4c,d). Furthermore, very short stalk lengths or conidiophore loss were frequently observed in the Δ*mpkB* mutant (Figure 4e). In the complemental strain of the *mpkB* deletion mutant, the abnormal conidiophore morphology was restored to the wild-type forms (Figure 4f). On the other hand, all strains (wild type and mutants) developed asexual structures in submerged culture. Conidiophores and conidia were observed in wild type (NRRL3357), the recipient strain (NRRL3357-5), and the Δ*mpkB* (SCWS1.11) mutant upon being cultured in MM broth at 30 °C and 250 rpm for 3 days. Particularly, the Δ*mpkB* mutant exhibited more than twice the production of conidiophores compared to the other strains (Figure 5).

### 3.4. Overexpression of mpkB Does Not Affect Growth and Development

Conditional overexpression mutants of *mpkB* were constructed using the promoter of the nitrite reductase gene *niiA*. However, the *mpkB* overexpression strain showed no significant difference with the wild type both in repressive and overexpressed conditions. When the strain was cultured in MM supplemented with sodium nitrate as a sole nitrogen source, hyphae growth, conidiation, conidiophore morphology, sclerotia production, and sclerotia size were not different from those of the wild type (Figure 6). Additionally, agar plate pigmentation and aflatoxin production were examined, but once again no significant differences were observed between the overexpression mutant and the wild type.

### 3.5. MpkB Regulates brlA Expression during the Developmental Stage

Northern blot analysis was performed to confirm whether the expression patterns of the major transcription factors and regulators associated with fungal development exhibited any differences in the Δ*mpkB* mutant (Figure 7). Expression of *brlA* increased continuously from 6 h to 36 h in the asexual induction conditions in the wild types, whereas its expression decreased under sexual induction conditions (see Materials and Methods). However, *brlA* expression in the Δ*mpkB* mutant remained unchanged in both asexual/sexual induction conditions. In the case of *nsdD* and *steA*, both transcripts were identified in the vegetative growth state of the Δ*mpkB*, but the expression pattern was largely similar to that of the wild type the rest of the time. The *veA* gene also exhibited no significant differences in expression levels and patterns between the wild type and the Δ*mpkB* mutant. *nsdC* was not detected by Northern blot analysis in either the wild type or the Δ*mpkB* mutant. Expression of *mpkB* in the wild type reached the highest expression level at the early stage of asexual development (6 h) and gradually decreased thereafter. In the sexual development induction conditions, the expression level of *mpkB* was less than that during asexual development, and *mpkB* transcripts were not detected in the vegetative growth stage.

### 3.6. MpkB Is Not Required for Aflatoxin Production in A. flavus

The production of aflatoxin by the wild type and mutant strains was measured by TLC. Interestingly, aflatoxin B1 production was confirmed in all strains including the Δ*mpkB* mutant (Figure 8). Moreover, no differences in aflatoxin yields were observed between the strains. Similarly, no significant differences in aflatoxin B1 production were observed between the wild-type and Δ*mpkB* mutants (both in mycelia and culture solution), as demonstrated by reverse-phase HPLC (Appendix A).

## 4. Discussion

The complete life cycle of *A. flavus* has been reassessed with the discovery of the sexual reproduction cycle associated with their sexual structure. However, the physiological and genetic regulatory mechanisms associated with the *A. flavus* life cycle remain unclear [14,15,47]. In our study, deletion of *mpkB* in *A. flavus* resulted in a marked decrease in conidiation, serious defects in asexual morphology. The abnormality of conidiophore morphology was very similar to that of *A. flavus* NsdC and NsdD mutants, which are homologs of *A. nidulans* NsdC and NsdD. Cary et al. [29] reported that *A. flavus* Δ*nsdC* and Δ*nsdD* mutants produced fewer conidia than the wild type and exhibited abnormal conidiophore morphology (e.g., short stalk and few phialides). Additionally, many studies have shown that yeast Fus3/Kss1 type MAP kinases are associated with conidiation in many filamentous fungi such as *Fusarium proliferatum*, *Fusarium graminearum*, *Botrytis cinerea*, *Alternaria alternata*, and *Cochliobolus heterostrophus* [48,49,50,51,52]. This result was also consistent with a study that characterized *mpkB* in *A. nidulans*, a model filamentous ascomycete. We have previously reported that *A. nidulans* MpkB MAP kinase affects asexual reproduction in this fungus, including conidia viability, germination, colony growth, conidiation, conidiophore morphology, and hydrolase gene expression [35,53]. Furthermore, the expression of *A. nidulans brlA* was inhibited by MpkB in the mid and late asexual development stage, as well as all sexual development stages, and therefore asexual development was regulated by MpkB MAP kinase [53]. Similarly, in *A. flavus, brlA* was found to be expressed not only in the asexual development stage but also in sexual development induction conditions (hypoxic and dark condition). Additionally, *brlA* transcripts in the Δ*mpkB* mutant were upregulated in the sexual development stage compared to the wild type. These results suggest that the expression of *brlA* was genetically regulated by *mpkB* in *A. flavus* in much the same way as in *A. nidulans*. To further confirm for this, we examined the number of conidiophores formed in the mycelial ball during submerged culture (i.e., conditions under which cell differentiation is generally inhibited, particularly in *A. nidulans*) and found that more conidiophores were produced in the Δ*mpkB* mutant than in the wild type. These results demonstrate that *A. flavus* MpkB MAP kinase signaling is highly involved in asexual reproduction and regulates the expression of *brlA*, an important transcription factor for asexual development. However, inhibition of *brlA* expression by MpkB in *A. flavus* is not absolute, as is the case in *A. nidulans*. Given that *A. nidulans* and *A. flavus* may have different balanced regulation systems with sexual and asexual developmental processes, the differences in the *brlA* expression pattern between *A. nidulans* and *A. flavus* might be related to the extremely low sexual reproduction frequency of *A. flavus* in nature. To address these questions, it is necessary to investigate the regulatory action of the MAP kinase signal of external stimuli activated-MpkB.

MpkB MAP kinase signaling in *A. flavus* can also regulate the production of sclerotia. The fact that yeast Fus3/Kss1 type MAP kinase is associated with the production of hyphal fusion or sclerotia has already been confirmed in several fungi species including *B. cinereal*, *Neurospora crassa*, *Fusarium oxysporum*, and *Sclerotinia sclerotiorum* [50,54,55,56]. In the case of *A. nidulans*, the Δ*mpkB* mutation appeared to block the production of cleistothecia and ascospores [35,36]. Our finding also revealed that *A. nidulans* Δ*mpkB* mutants failed to carry out anastomosis to form heterokaryotic hyphae in the early stage of the sexual development process; therefore, this mutant could not form a zygote structure [35]. In *A. flavus*, higher cell densities corresponded with lower sclerotia production and increases in conidiation and vice versa. *A. flavus* is known to balance its conidiation and sclerotium differentiation process via cell density-dependent mechanisms [57]. Cell–cell interaction or anastomosis is thought to regulate conidiation/sclerotium differentiation in *A. flavus* in a cell density-dependent manner, a process in which MpkB MAP kinase signaling might be involved. Previous studies have revealed that the asclerotium phenotype of *A. flavus* appeared when regulators or transcription factors related to sexual development and secondary metabolism of the fungus (e.g., *veA*, *velB*, *laeA*, *nsdC* and *nsdD*) were not functional [22,24,29,58]. The fact that *A. flavus* ascocarps are produced inside the sclerotium and that the production of sclerotia is regulated by MpkB MAP kinase suggests that sclerotia are neither merely an overwintering survival structure nor an ascocarp reserve, but also a necessary sexual organ. In the absence or the inactivation of MpkB protein in *A. flavus*, heterothallic mating is not expected to form sexual sclerotia, or produce a zygote structure for ascocarp formation.

The most interesting feature of the *A. flavus* Δ*mpkB* mutant is that it synthesizes aflatoxin B1 normally. The development and secondary metabolism of filamentous ascomycetes are generally considered to be genetically linked and co-regulated [59]. In *A. nidulans* studies, MpkB MAP kinase signaling was linked to the synthesis of secondary metabolites sterigmatocystin, penicillin, and terrequinone A [34,37]. Activated MpkB MAP kinase in *A. nidulans* phosphorylates VeA, an important regulatory protein for sexual development and secondary metabolite production, and Δ*mpkB* mutants exhibited a reduction in velvet protein complex formation compared to wild type [37]. Many studies have shown that the synthesis of secondary metabolites and the sclerotial production of *A. flavus* and *A. parasiticus* are also accurately controlled by the VeA activator and LaeA regulator [21,22,24]. The Δ*nsdC* and Δ*nsdD* mutants of *A. flavus* also exhibit a blocked or markedly reduced aflatoxin synthesis, and the expression of *aflR* is increased in mutants via the deletion of these two nsd genes [29]. On the other hand, the deletion of *fluG* in *A. flavus* resulted in delayed or decreased conidiation compared with the wild type, which was coupled with an increase in sclerotia production; however, the aflatoxin synthesis capacity of the Δ*fluG* strain remained normal [60]. The differences between *A. flavus* and *A. nidulans* Δ*fluG* strains regarding their ability to synthesize aflatoxin or sterigmatocystin indicate that the function of the FluG protein in these two species is different. These observations suggest that these two species of *Aspergilli* possess a conserved and divergent signaling pathways associated with the regulation of asexual sporulation and secondary metabolism. Interestingly, according to Frawley et al. [39], an *A. flavus mpkB* knockout strain did not produce aflatoxin. Not only the ∆*mpkB* mutant but also mutations in the components of the MpkB module resulted in a lack of aflatoxin production. In contrast, other secondary metabolites in the ∆*mpkB* exhibited higher yields than those of the wild-type strain. Moreover, in addition to the aforementioned effects on aflatoxin production, the mutant developmental phenotypes were consistent with our results, implying that the developmental processes governed by MpkB signaling and modulation of secondary metabolism could be diverse. This phenotypic difference might be due to differences in strain backgrounds. To address this, we constructed an *A. flavus* knockout mutant of the *mkkB* gene (AFLA_103480), an upstream MAPK kinase gene of *mpkB*, using the same host strain (NRRL3357-5). Production of aflatoxin was reverified in the ∆*mkkB* mutant, thereby confirming the same aflatoxin production as in the wild-type and the ∆*mpkB* mutant These results concerning *mkkB* will be published in a future study. *A. flavus* populations are known to exhibit a diversity of morphologies, mycotoxin production, and vegetative compatibility groups (VCGs). Particularly, secondary metabolite profiles vary depending on the *A. flavus* population VCG, which is also true for aflatoxin B1 and B2 yields [61]. Given that the high genetic diversity of these fungal groups can result in morphological differences and variations in their ability to produce secondary metabolites [3,4], we speculate that the differences in the genetic background of the *mpkB* deletion mutants might result in different secondary metabolites profiles.

## 5. Conclusions

Our study confirmed the function of the *A. flavus* MpkB MAP kinase, an ortholog of the *S. cerevisiae* Fus3/Kss1 that is associated with conidiation, conidiophore morphogenesis, sclerotiogenesis, and aflatoxin biosynthesis. Furthermore, our findings suggest that MpkB MAP kinase signaling regulates the conidiation of *A. flavus* via repression of *brlA* expression, although the regulation of *brlA* expression is not as consistent and strong as that of *A. nidulans*. Conidiophore morphogenesis also requires MpkB. Specifically, the reduction of conidia production in the *A. flavus* Δ*mpkB* mutant was presumably due to the abnormal conformation of the conidiophore. MpkB MAP kinase has also been shown to be essential for sclerotia production. However, MpkB was not required for the biosynthesis of aflatoxin. All of these results lead us to speculate that MpkB MAP kinase signaling is directly or indirectly involved in the development and regulation of secondary metabolism in *A. flavus*, including velvet complex formation. However, there is currently no empirical evidence to support this hypothesis, and therefore more studies are required to further clarify the role of MpkB.

## Figures and Tables

**Figure 1 jof-06-00289-f001:**
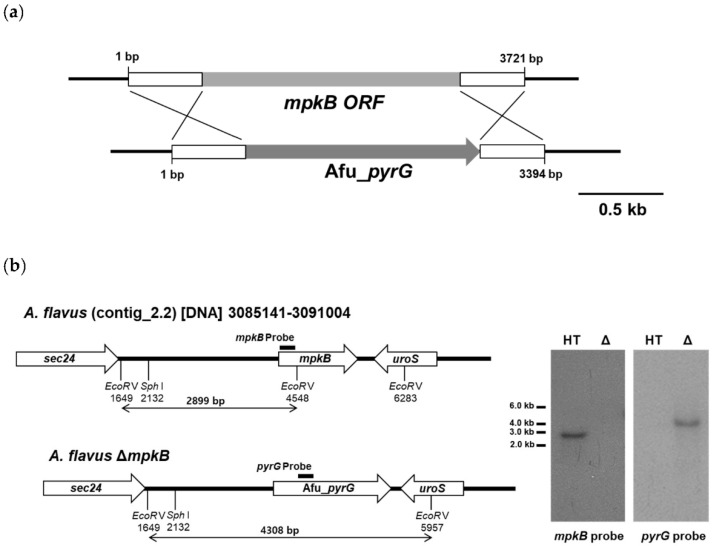
Deletion of the *mpkB* gene in *A. flavus* NRRL3357-5 (*pyrG*^−^). (**a**), Schematic model of the approach used to disrupt the *mpkB* gene using the *A. fumigatus pyrG* marker gene. The open boxes represent the flanking regions of the *mpkB* gene. The 1 kb upstream and the 0.8 kb downstream flanking regions of the *mpkB* gene were respectively attached at the 5′- and 3′-ends of the knockout cassette to introduce direct repeat sequences. (**b**), Southern blot analysis of the *mpkB* deletion mutant. Genomic DNA was digested with *EcoR*V, separated via agarose gel electrophoresis, and subjected to Southern blot analysis, after which we identified a clone that exhibited the expected banding pattern associated with *mpkB* deletion. HT and Δ represent the recipient strain for transformation (NRRL3357-5) and the gene disruptant (SCWS1.11), respectively.

**Figure 2 jof-06-00289-f002:**
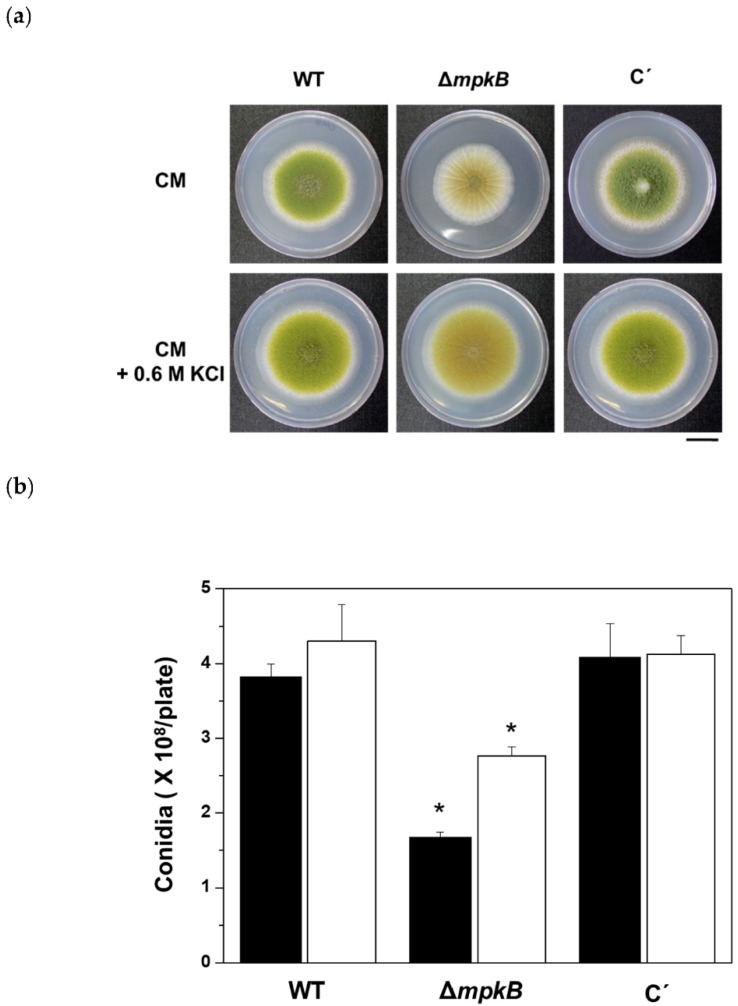
Growth and asexual sporulation of the *mpkB* deletion mutant. (**a**), Wild type (NRRL3357), Δ*mpkB* (SCWS1.11) and *mpkB* complementation (C’) strain (SCWS1.20) were point-inoculated on CM agar and CM agar containing 0.6 M potassium chloride and incubated at 30 °C for 4 days. The scale bar indicates a 1.5 cm length. (**b**), Conidial quantitative analysis of the strains shown in (**a**). Black bars: CM plate counts; Open bars: CM + 0.6 M KCl plate counts. The conidia were counted three times for each plate. The error bars indicate standard deviations (* *p* < 0.05).

**Figure 3 jof-06-00289-f003:**
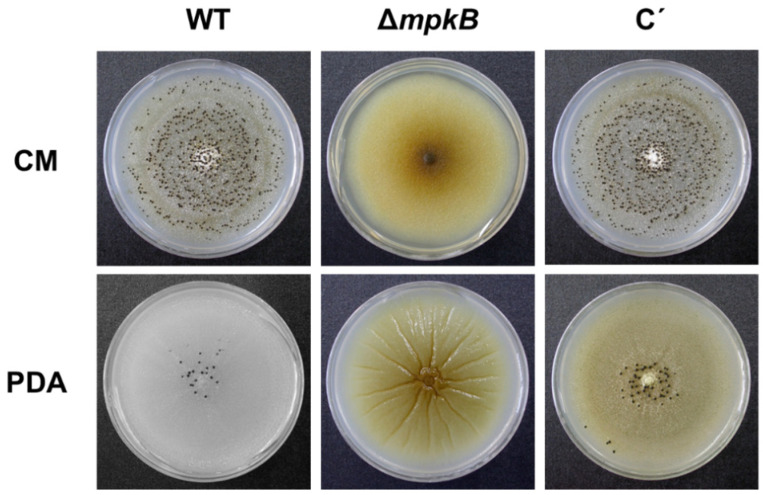
Sclerotial production of the *A. flavus mpkB* deletion mutant in CM agar and PDA plates. The images were captured from the plates that were cultured for 6 days at 30 °C in dark conditions. The conidia produced by the colonies were washed off with 70% ethanol to expose the sclerotia. The wild-type (WT) strain was *A. flavus* NRRL3357, the mutant (Δ*mpkB*) is SCWS1.11, and the complemental strain (C’) is SCWS1.20.

**Figure 4 jof-06-00289-f004:**
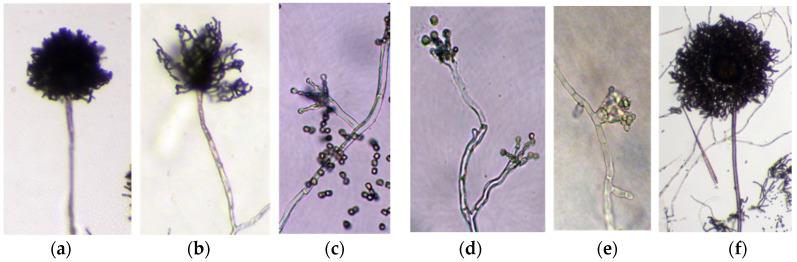
Conidiophore morphological phenotypes of the *mpkB* deletion mutant. (**a**), Conidiophore of the wild-type strain (NRRL3357). (**b**–**e**), Conidiophore of the Δ*mpkB* (SCWS1.11) mutant strain. (**f**), Conidiophore of the complemental strain (SCWS1.20). The conidia of each strain were harvested from the CM plates after a 4 day incubation period and inoculated in mounted CM agar blocks. All images were captured from cultures growing for 3 days at 30 °C. (**a**,**b**,**f**) are magnified at 100×, the remaining images are at 400×.

**Figure 5 jof-06-00289-f005:**
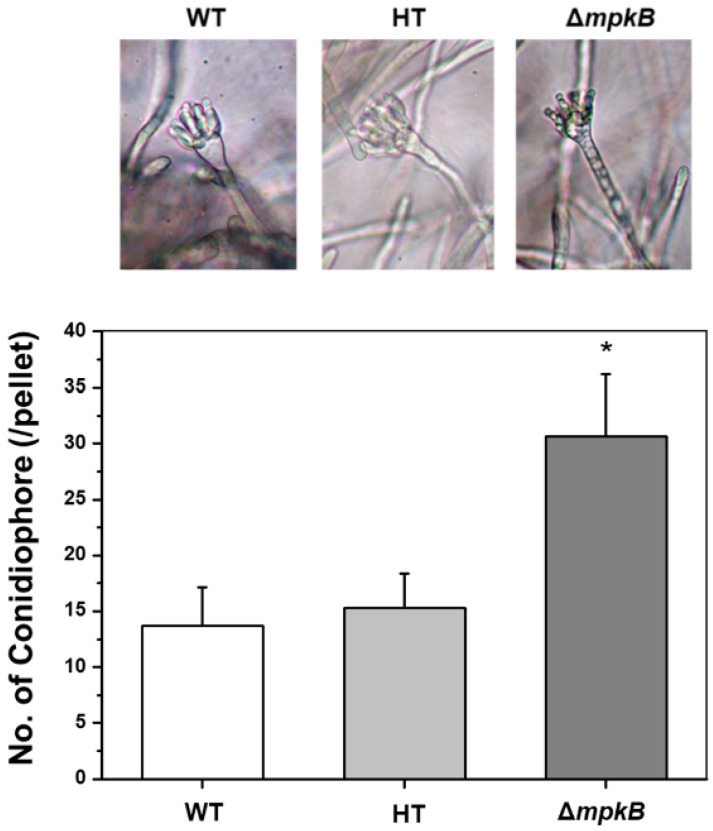
Characterization of *A. flavus* strains in submerged culture. Conidiophores were produced in all strains. The magnification of the images is 400×. 1 × 10^−6^ conidia were inoculated and grown in MM broth for 3 days at 30 °C and 250 rpm in the light. The numbers of conidiophores for each pellet were counted on a dissecting microscope. WT: wild type (NRRL3357), HT: recipient strain (NRRL3357-5), Δ*mpkB*: *mpkB* deletion mutant (SCWS1.11). The error bars indicate standard deviations (* *p* < 0.05).

**Figure 6 jof-06-00289-f006:**
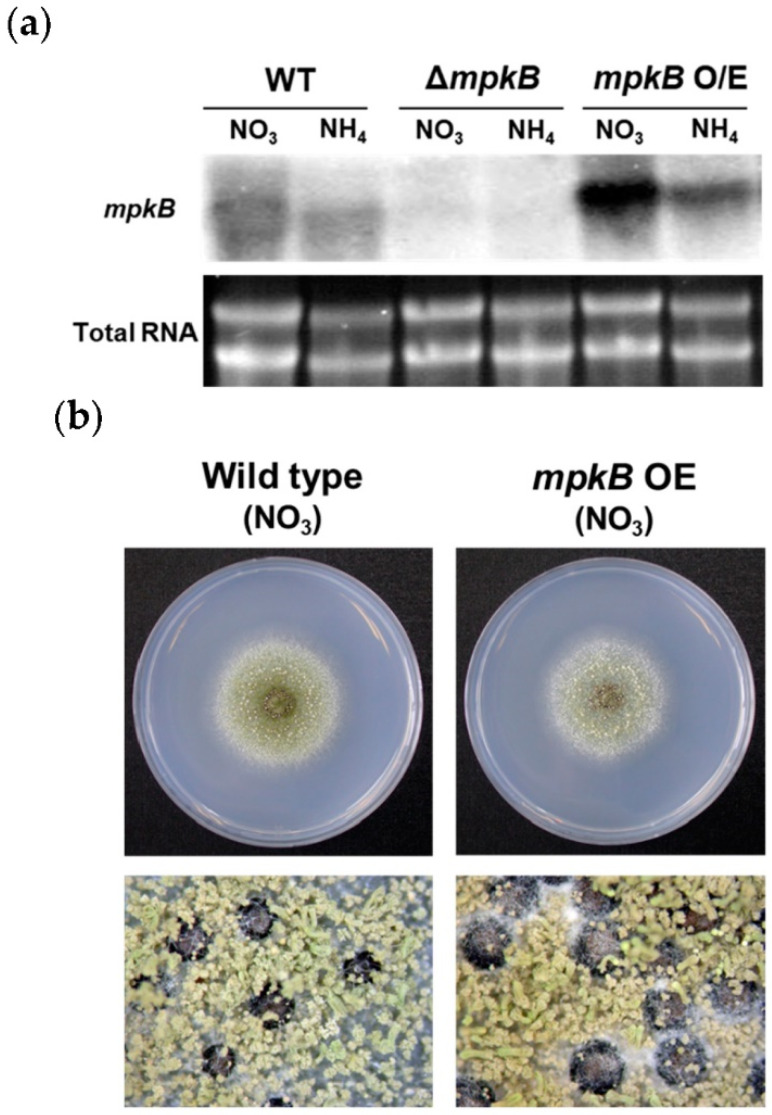
Overexpression of the *mpkB* allele in the *A. flavus* NRRL3357-5 strain. (**a**) Expression levels of *mpkB* mRNA in the NRRL3357 (WT), SCWS11.1 (Δ*mpkB*), and SCWS1.35 (*mpkB* O/E) strains. Mycelia were grown for 16 h in CM broth before being transferred to solid MM plates. The plates were further incubated to 18 h with *niiA*(p)-inducible (containing NO_3_) or repressible conditions (containing NH_4_). Equal amounts of total RNA were evaluated via ethidium bromide staining. (**b**) NRRL3357 (WT) and SCWS1.35 (*mpkB* O/E) were point-inoculated onto *niiA*(p)-inducible media and incubated for 4 days at 30 °C (upper). The bottom photographs are 6 day cultures (37.5× magnification).

**Figure 7 jof-06-00289-f007:**
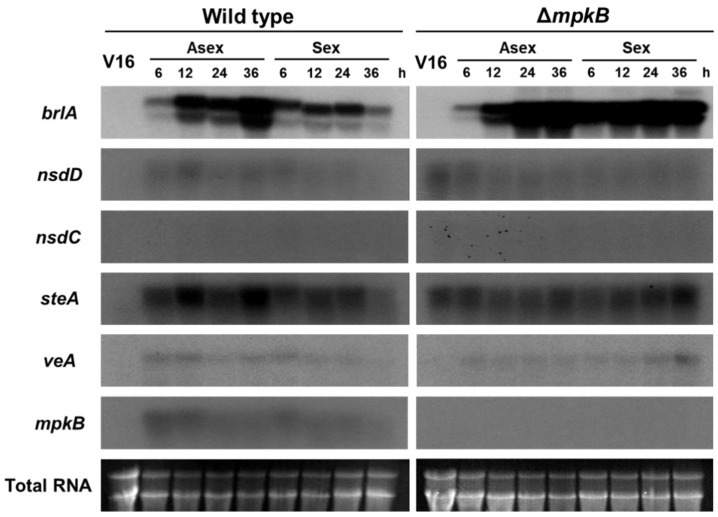
mRNA expression of *brlA*, *nsdD*, *nsdC*, *steA*, *veA*, and *mpkB* in *A. flavus* NRRL3357 and Δ*mpkB* (SCWS1.11). NRRL3357 and SCWS1.11 were grown in CM broth with constant shaking (250 rpm) at 30 °C for 16 h, after which the mycelium balls were transferred to solid MM plates for asexual or sexual development. Asexual development was induced by aerating the cultures and keeping them at 30 °C. For sexual induction, after 24 h of incubation under hypoxic and dark conditions in sealed plates, the seals were removed, and the cultures were incubated further. Northern blots were probed with open reading frame (ORF) amplicons. “V16” indicates the vegetative growth stage at 16 h. “Asex” and “Sex” indicate asexual and sexual development inducing conditions, respectively.

**Figure 8 jof-06-00289-f008:**
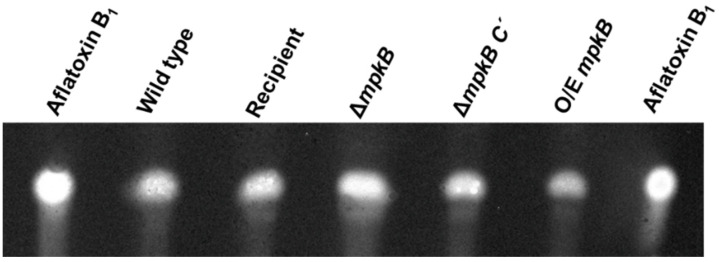
TLC analyses of aflatoxin production by the *A. flavus* wild type, Δ*mpkB*, and Δ*mpkB* complemental strains, as well as the *mpkB* overexpression mutants. The strains were grown in potato dextrose broth for 3 days at 30 °C and extracts were separated on a TLC plate. Extract samples from each strain were assessed as stationary cultures. Aflatoxin was visualized using a UV (365 nm) light.

**Table 1 jof-06-00289-t001:** *A. flavus* strains used in this study.

Strain	Genotype	Use	Reference
NRRL3357	Wild type	Wild-type strain	[40]
NRRL3357-5	*pyrG^−^*	Recipient strain	[40]
SCWS1.02	*pyrG^−^*, *Afu_pyrG*	*pyrG* complementation strain	This study
SCWS1.11	*pyrG^−^*, Δ*mpkB:Afu_pyrG*	*mpkB* deletion strain	This study
SCWS1.20	Δ*mpkB::Afu_pyrG; mpkB, ptrA*	*mpkB* complementation strain	This study
SCWS1.35	*pyrG^−^; niiA*(p)*::mpkB, Afu_pyrG*	*mpkB* overexpression strain	This study

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
