# Peer review of "The Conserved MAP Kinase MpkB Regulates Development and Sporulation without Affecting Aflatoxin Biosynthesis in Aspergillus flavus"

_jof, 2020, doi:10.3390/jof6040289_

Round 1

Reviewer 1 Report

The paper has drastically improved.

Some minor comments no deal with italics of species and gene names and latin plural forms of terms:

L38: sclerotia is the plural of sclerotium

L79 cleistothecia

L202 should mpkB not appear in italics?

L205 A. nidulans certainly should

L222 FUS3 in italics

L223 A. flavus in italics

L314 “The complete generation of A. flavus, a heterothallic fungus, has been reassessed” : rephrase; life cycle is meant?

L325 conidia

The discussion repeats the results. This should be avoided and rewritten.

This can be accomodated in a minor revision.

Author Response

Reviewer 1

The paper has drastically improved.

Some minor comments no deal with italics of species and gene names and latin plural forms of terms:

Response:  Thank you for the careful comments and suggestions. We corrected as your suggestions.

L38: sclerotia is the plural of sclerotium

Response: L38: corrected with sclerotia.

L79 cleistothecia

Response: L78: corrected with cleistothecia.

L202 should mpkB not appear in italics?

Response: During the correction of the sentence, there was an error in the word that should be italicized from L184 to L299. All corrected.

L205 A. nidulans certainly should

Response: Corrected.

L222 FUS3 in italics

Response: Corrected.

L223 A. flavus in italics

Response: Corrected.

L314 “The complete generation of A. flavus, a heterothallic fungus, has been reassessed” : rephrase; life cycle is meant?

Response: Yes, it meant life cycle. To make clear, we changed the sentence as follows;

‘The complete life cycle of A. flavus has been reassessed with the discovery of the sexual reproduction cycle associated with their sexual structure.’

L325 conidia

Response: L331: Corrected with conidia

The discussion repeats the results. This should be avoided and rewritten.

Response: Some repeated part of the discussion was revised and shortened.

Reviewer 2 Report

Dear Authors

The article deserves publishing, only minor changes and implementation are requested. See the attached file.

Regards

Author Response

Point 1: In the text 50% was reported

Response 1: Thank you for the careful comment. However, our text does not report 50%. Please check again.

Point 2: add the composition of MM and CM or insert a reference.

Insert potato dextrose broth (PDB)

Response 2: L105-L109: We corrected as your suggestions.

Point 3: this is the initial concentration? if yes, what is the final concentration?

Response 3: L109: presented in the text is final concentration.

Point 4: with the addition of uridine and uracil

Response 4: L152-L153: We corrected as your suggestions.

Point 5: please add 'agar'

Response 5: L159: corrected.

Point 6: delete agar

Response 6: L159: corrected.

Point 7: delete PDA broth and insert PDB

Response 7: L174: corrected.

Point 8: rpm?

Response 8: L177: rpm value was inserted.

Point 9: Agar

Response 9: L209: corrected.

Point 10: Agar

Response 10: L210: corrected.

Point 11: insert this method in materials and methods

Response 11: L211-L213: We corrected as your suggestions.

Point 12: CM agar

Response 12: L224: corrected.

Point 13: -6?

Response 13: L263: corrected with 1x10-6

Point 14: the agar plate pigmentation was also evaluated with TLC???

Response 14: L274: Corrected the sentence.

This manuscript is a resubmission of an earlier submission. The following is a list of the peer review reports and author responses from that submission.

Round 1

Reviewer 1 Report

Review on

I found an abstract from 2015 on: https://www.aspergillus.org.uk/conference_abstracts/the-conserved-map-kinase-mpkb-affects-developmental-processes-but-not-secondary-metabolism-in-aspergillus-flavus/

Stating “The conserved MAP kinase MpkB affects developmental processes but not secondary metabolism in Aspergillus flavus” with another Co-author: Kwang-Yeop Jahng. Why was he not included in this paper.

I wonder what took five years to produce this manuscript…

The paper analysis the mpkB homolog of A. nidulans in A. flavus. This is an important study although the expected results are similar to the A. nidulans mutant. This should be compared e.g. in a table to highlight any novel result.

Key findings: deletion of AflmpkB resulted in no mycelial growth change, while the conidial production was reduced about 60% comparing to the wild-type. The mutant produced immature and abnormal conidiophores, decreased number of the phialides and very short stalks. Expression of brlA was up-regulated in the mutant. ΔAflmpkB did not produce any sclerotia. However, AflmpkB mutants produced normal level of aflatoxin B1.

L13: genes should be in italics, here FUS3 check elsewhere in the manuscript.

L21+ correct last sentence of the abstract.

L26+ there is no such thing as a “phylogenetic strain” – correct next sentence

L30 since when are peanuts grains?

L32 there is no such thing as a polyketide-driven … synthesis is based on… are the most important carcinogenS

L33+ correct sentence

L38 sclerotia correct this sentence, it is ill-formulated

L39+ correct this sentence and the next as well…

This paper is unreadable and I stop reviewing here..

L331: misspelled: Botrytis cinerea

Figure 1: change “Coli ori” in ColE1ori or E. coli ori